# Behavioural Synchronisation between Dogs and Humans: Unveiling Interspecific Motor Resonance?

**DOI:** 10.3390/ani14040548

**Published:** 2024-02-07

**Authors:** Angélique Lamontagne, Florence Gaunet

**Affiliations:** 1Centre de Recherche en Psychologie et Neuroscience (UMR 7077), Aix-Marseille University, Centre National de la Recherche Scientifique, 3 Place Victor Hugo, 13331 Marseille, Cedex 03, France; 2Association Agir pour la Vie Animale (AVA), 76220 Cuy-Saint-Fiacre, France

**Keywords:** dog, human–animal interaction, behaviour, synchronisation, motor resonance, mirror neurons, motor contagion, interspecific cognition

## Abstract

**Simple Summary:**

It is widely acknowledged that dogs synchronise their behaviours with those of humans. In this article, we delve into the neural and cognitive bases of this form of behavioural adjustment. Using knowledge of cerebral processes underlying behavioural synchronisation in humans, namely motor resonance and the activation of mirror neurons, we investigated whether dogs’ behavioural synchronisation with humans could be based on similar mechanisms. In humans, both perceiving and executing an action activate the motor representations of that action. Motor resonance occurs when motor representations of an action are simultaneously activated in both the observer and the individual being observed. Mirror neurons are the neural substrate for motor resonance as these motor neurons activate both when an individual performs or perceives an action. Our review of existing literature shows that motor resonance could occur in dogs, suggesting that dogs’ behavioural synchronisation with humans relies on the activation of both human and canine mirror neurons. Behavioural studies suggest but do not prove the existence of motor resonance between the two species, and thus further research is needed to confirm this hypothesis. Revealing the mechanisms underlying dogs’ behavioural synchronisation with humans would contribute to a better understanding of how dogs and humans have evolved together since the beginning of their cohabitation.

**Abstract:**

Dogs’ behavioural synchronisation with humans is of growing scientific interest. However, studies lack a comprehensive exploration of the neurocognitive foundations of this social cognitive ability. Drawing parallels from the mechanisms underlying behavioural synchronisation in humans, specifically motor resonance and the recruitment of mirror neurons, we hypothesise that dogs’ behavioural synchronisation with humans is underpinned by a similar mechanism, namely interspecific motor resonance. Based on a literature review, we argue that dogs possess the prerequisites for motor resonance, and we suggest that interspecific behavioural synchronisation relies on the activation of both human and canine mirror neurons. Furthermore, interspecific behavioural studies highlight certain characteristics of motor resonance, including motor contagion and its social modulators. While these findings strongly suggest the potential existence of interspecific motor resonance, direct proof remains to be established. Our analysis thus paves the way for future research to confirm the existence of interspecific motor resonance as the neurocognitive foundation for interspecific behavioural synchronisation. Unravelling the neurocognitive mechanisms underlying this behavioural adjustment holds profound implications for understanding the evolutionary dynamics of dogs alongside humans and improving the day-to-day management of dog–human interactions.

## 1. Introduction

Behavioural synchronisation occurs when individuals exhibit the same behaviour in the same location and at the same time. This form of behavioural alignment is widely documented and ubiquitous in social species (see [1,2] for comprehensive reviews). Dogs, like many social species, exhibit behavioural synchronisation at the intraspecific level. During group activities, such as play sessions or when moving in a pack, dogs copy the behaviour of their peers and behave synchronously by staying close to each other, exchanging glances and vocalising [3,4,5]. These behaviours are the cornerstone of dogs’ social interactions. Dogs, however, hold a unique and significant role in human history, marking the earliest instance of domestication, over 20,000 years ago [6,7,8,9,10]. The process of domestication involved the close coexistence of wolves, the ancestors of dogs, and humans, sharing territory and living closer and closer together over time [11]. These wolves were predators and ate the same prey as humans. Selected wolves were fed and raised by humans and became increasingly dependent on humans [12]. They then assisted humans in tasks such as hunting livestock protection. Archaeological evidence indeed reveals that through domestication, dogs maintained close contact with humans, living near them, initially scavenging human food remains and later participating in hunting expeditions [13,14]. Genomic analyses provide compelling evidence of the enduring and close relationship between dogs and humans [6,7,10,15,16,17]. These analyses reveal that dogs often accompanied groups of humans on their journeys, although instances exist where humans ventured without their canine companions. Dogs traversed between human communities, or their presence was a result of cultural and economic exchanges [6]. This shared history has entrenched dogs as ubiquitous members of human communities worldwide, serving as companions, working partners, or cohabitants within villages [16]. Today, dogs have indeed become ubiquitous in human environments; they are able to adapt socially and spatially to urban living [18,19]. In many societies, dogs are no longer considered solely as working animals but also as members of households, sharing many daily activities alongside humans [14,19]. The millennia that dogs and humans have spent in cohabitation have led dogs to develop specific cognitive capabilities [20,21]. Notably, dogs are sensitive to human behavioural cues, following the direction of humans’ gazes or pointing gestures. They are also capable of referential communication with humans [22,23,24,25,26]. More recently, studies have revealed that dogs synchronise their behaviour with that of humans, using the example of locomotor synchronisation, i.e., the synchronisation of locomotor behaviours. A key part of human–dog interactions involves dogs walking alongside humans: humans and dogs walk together in many everyday situations [19,27,28]. In urban settings, dog owners take their dogs on two to four daily walks around their neighbourhood and walk or run with their canine companions; during their walks, dogs and owners mutually adjust their walking pace to each other. Experimental studies have demonstrated that dogs spontaneously synchronise their locomotor behaviour with their owners during off-leash walks without explicit reinforcement, whether indoors or outdoors, in known or unknown environments [20,29,30,31,32]. Interspecific behavioural synchronisation between dogs and humans is thus well documented. However, the existing body of research has focused on demonstrating this form of behavioural alignment but has not delved into its underlying neurocognitive mechanisms. Given the distinctive cognitive abilities observed in dogs, we can reasonably assume that behavioural synchronisation between dogs and humans shares similar neurocognitive underpinnings with that observed in human–human interactions. In humans, behavioural synchronisation hinges on motor resonance, i.e., the simultaneous activation of motor representations in interacting individuals (see [33] for a recent and emblematic review). Motor resonance is linked to action perception coupling and is associated with the activation of mirror neurons. The current review aims to offer a comprehensive examination of the existing literature on the underlying mechanisms of behavioural synchronisation, in an effort to determine whether action perception coupling and mirror neurons might serve as the neurocognitive underpinnings for dog–human behavioural synchronisation. Furthermore, we will explore the properties of motor resonance, including motor contagion and its modulation, to ascertain the consistency of findings in interspecific studies with these fundamental properties. This review represents the first exploration of the mechanisms underlying interspecific behavioural synchronisation, ultimately contributing to our understanding of the intricate co-evolutionary relationship between dogs and humans.

## 2. Sensorimotor Prerequisites of Motor Resonance

### 2.1. The Case of Humans

Humans, like other social species, tend to align their behaviours with one another (see [34] for a review on behavioural alignment in humans). Behavioural synchronisation is a form of behavioural alignment, like mimicry and automatic imitation. Neurological studies conducted in humans found that behavioural alignments rely on motor resonance [35,36,37,38,39]. The concept of motor resonance originates from the field of physics and is used to describe the phenomenon in which one system oscillates at the same frequency and in phase with another system [40]. When applied to social interactions, motor resonance occurs when the motor systems of interacting individuals are in phase. Specifically, when individuals mutually align their behaviours, the sensorimotor representations associated with these behaviours are activated simultaneously in the brain of each interacting individual [40].

As far as we know, motor resonance and its prerequisites have only been studied in humans. To achieve motor resonance, people must be capable of perceiving the behaviour of others. In other words, perceiving others’ behaviours is necessary for motor resonance to occur. The perception of another person’s action activates sensory representations associated with that action in the observer. These sensory representations then activate motor representations related to the observed action [41]. In other words, sensory perception is translated into an internal representation in the observer’s brain, first sensory, and then motor [42]. This activation is triggered through the Action Observation Network (AON), which encodes observed actions into the motor program (see Figure 1) [43]. This network automatically activates when perceiving an action. With this network, motor representations are activated following action perception, and thus, motor resonance occurs, making observers more inclined to perform the action themselves [35,44,45]. The perception of an action thus directly influences the motor response. Furthermore, the perception of action and the production of action share a common representational format. The correspondence between perceived and executed actions corresponds to action perception coupling [42]. Perception and action are thus not distinct and independent processes but are interconnected and interdependent (see Figure 1) [45].

Hence, when individuals interact, a system is in place to detect social alignment or misalignment, and this process relies on the perception of the other’s behaviour [46]. A two-stage process then comes into play (see Figure 1) [47,48]. The first stage is a bottom-up process: the perception of the behaviour induces a corresponding state in the observer, leading to motor resonance as described above. This first stage prompts a rapid response characterised by a tendency to reproduce the perceived behaviour [47]. This early response was demonstrated in a transcranial magnetic stimulation study, revealing a stimulus-driven mirror response in the subjects’ motor cortex within 150 ms of the onset of the observed behaviour [49]. Following this first stage, top-down processes reinforce or inhibit the initial response. These processes are contingent upon various factors related to the observer and the people initially producing the behaviour (see [2] for an exhaustive review of these social factors). This leads to another, later motor cortex response, around 300 ms after the behaviour is perceived [45]. If this second stage reinforces the initial response, then motor resonance is not inhibited, resulting in the observer performing the perceived behaviour [41]. This behavioural response triggers brain reward circuits, strengthening the mutual adjustment between individuals and enhancing their social relationship [46]. Being able to perceive and decode the behaviours of others is thus a prerequisite of motor resonance and allows one to adapt one’s own behavioural response [50]. Note that this ability can be impaired, as in the case of ideomotor apraxia, for example. This neuropsychological disorder is caused by a cortical lesion, characterised by a loss of the ability to perform a gesture on demand or to imitate a perceived action [51].

The importance of perception in achieving motor resonance is emphasised by the fact that motor resonance depends on individuals’ level of attention. When people are fully attentive to each other, motor resonance occurs; if they are not entirely attentive to each other, motor resonance is reduced [43]. In a neurophysiological study, the amplitude of subjects’ motor responses was lower when they were instructed to simultaneously monitor the activity of an LED light while observing rhythmic hand movements, compared to a condition in which subjects solely observed the hand movements [43]. Behavioural studies also demonstrate that a certain level of attention between the interacting partners is necessary to achieve behavioural synchronisation [52,53,54]. For example, individuals seated in rocking chairs exhibited less synchronisation in their rocking movements when positioned side by side, when they were unable to see each other, in contrast to when they were face to face and could be visually attentive to each other [55]. Attention toward a social agent is thus a prerequisite for motor resonance to occur.

### 2.2. Are Prerequisites of Motor Resonance Present in Dogs?

Individuals need to be able to perceive and decipher the actions of others to experience motor resonance. In social species, individuals are able to discriminate the behaviours of their congeners; this capacity is necessary for communication. Like all social species, dogs have the ability to discriminate the behaviours of their conspecifics and take this information into account to adjust their own behaviours [56,57,58]. During pack conflicts, for instance, dogs observe their packmates, which is interpreted as a means to recruit them for combat or to coordinate their movements for synchronised collective action [56]. This ability is also observed when dogs hunt. Hunting in canids is based on the transfer of information between individuals during group movements [59,60] and as observed in African Wild Dogs, it implies that individuals focus their attention on a single prey and relate to one another in space and time [61,62]. Dogs, like other canids, are able to identify the position and activity of their partners in order to hunt cooperatively. When pursuing prey, a few individuals hide in ambush, waiting for other pack members to bring the prey toward them. A relay run can also occur during pursuit; this is a type of cooperative running in which canids alternately play different roles in the hunting activity [63]. The motor sequence of canids hunting is triggered by the movement of the prey, after which canids display specific behaviours: they move toward the prey, then pursue it by fanning out, approaching it, and circling it until it becomes motionless (see Figure 2) [63].

Dogs distinguish not only the behaviours exhibited by other dogs but also those performed by humans. The process of domestication led to the selection of specific canine cognitive abilities toward humans, a process which has been the subject of numerous studies in recent decades. It is widely accepted that dogs perceive and copy some human behaviours. For instance, studies using the “Do as I do” procedure showed that dogs learned to copy actions demonstrated by humans such as turning around or jumping over an obstacle and then could generalise it to other human behaviours they had not specifically been taught, like opening a door or moving an object [64,65]. Dogs were also found to copy human actions even when they were not causally relevant: dogs copied their owner pushing a door leading to a food reward, but they also copied their owner touching coloured dots next to the door, even though it did not give them access to the food [20]. Lastly, another study showed that when faced with a stranger, dogs copied their owner’s behaviour, by approaching the person or staying back based on their owner’s behaviour [66]. Dogs are sensitive to human behavioural cues: they follow gaze direction, pointing gestures, and head orientations [21,67,68] and discriminate facial expressions [69] and body postures [50,70]. Dogs adjust their behavioural responses based on perceived human cues [20,26,71,72]. For example, they perform better at reaching a reward hidden behind a fence when the action is first demonstrated by a human than when there is no human demonstration [73]. Similarly, when a person hides food under one of several containers out of sight of the dogs and then points to indicate the location of the hidden treat, dogs follow the indicated direction to find the food [74]. Dogs are also capable of adapting their behavioural responses according to the human’s attentional state. In an experiment, dogs were presented with a visible and accessible food reward, but an experimenter prohibited them from touching the food [75]. The experimenter then either continued to look at the dog, left the room, turned their back, engaged in a distracting activity, or closed their eyes. Dogs’ behaviours varied depending on the person’s attentional state. Notably, the dogs approached the food less, disobeyed less frequently, and sat more often when the experimenter was looking at them compared to the other conditions. In another experiment, dogs were placed in a room and could see an inaccessible reward. They then positioned themselves optimally in relation to the height of the reward and the owner’s line of sight [23].

When dogs are presented with new situations, they seek information from their owners and adjust their behavioural responses based on perceived behaviours; this is known as social referencing [76,77]. These studies show that when faced with an unfamiliar object, dogs looked at their owner in a referential manner and adjusted their behaviour based on their owner’s reaction toward the object. Dogs approached the object more quickly when their owner had a positive facial expression compared to when the owner displayed a negative facial expression toward the object. Social referencing, initially studied in human dyads of young children and their mothers, thus also occurs in dog–owner dyads. Another capacity that was previously believed to be uniquely human and that has been observed during dog–human interactions is overimitation [78,79,80]. This behaviour occurs when an individual copies another individual’s action, even when the action is causally irrelevant. Overimitation has not been observed in other primate species and until it was observed in dogs during dog–human interactions, only humans were thought to be capable of copying irrelevant or functionally unnecessary actions demonstrated by a conspecific. Because dogs possess all these cognitive processes, they thus also possess the prerequisites for motor resonance in their interactions with humans. Dogs discriminate human behaviours, are sensitive to their attentional state, and adapt their motor responses based on perceived human social cues. They adjust their behaviours to those of humans, both in familiar contexts and in novel situations. It therefore seems reasonable to assume that behavioural synchronisation between dogs and humans is based on a mechanism akin to that occurring during human interactions, namely interspecific motor resonance.

## 3. Mirror Neurons, the Neuronal Substrate of Motor Resonance

### 3.1. Mirror Neurons and Interbrain Synchronisation in Primates

Mirror neurons were discovered three decades ago in the F5 area of the premotor cortex in monkeys [81,82,83,84]. Neurons homologous to those identified in the F5 region of primates were subsequently found in humans, notably in Brodman’s area, the superior parietal lobe, the inferior parietal lobe, the dorsolateral prefrontal cortex, the parietal cortex, and the premotor cortex [35,85,86,87]. The distinctive feature of mirror neurons is their activation both when a subject performs an action and when the subject observes another individual performing the same action [88]. Initially, mirror neurons were thought to respond exclusively to goal-directed actions [35,84]. Subsequent studies have identified additional properties of mirror neurons (see [89] for a review). Mirror neurons can respond to both observed actions and visually presented objects, to actions performed with a tool, and to nonbiological objects in motion. In monkeys, for example, cells in the lateral intraparietal area are activated both when the monkey looks in the preferred direction of the neuron and when another monkey looks in the same direction [90]. There are also canonical neurons in the ventral premotor cortex of monkeys that activate when the individual performs an action directed at an object and when the individual perceives the object as the target of the action [91]. Mirror neurons are therefore sensorimotor neurons with both motor and perceptual properties [91]. These neurons provide neurophysiological support to action perception coupling, as they fire during both the perception and execution of actions, proving the interconnection between motor and sensory areas [35,82,84,92]. When an individual observes a conspecific performing an action, the observer’s mirror neuron activity reflects the mirror neuron activity of the individual executing the action [91]. Mirror neurons are thus the neural underpinning for motor resonance. Neurophysiological studies support the correspondence of brain activity between interacting individuals. In early neurophysiological studies, the measurement techniques did not allow researchers to simultaneously record brain activity in multiple subjects during social interactions [86,93,94]. These studies were limited to single-brain recordings and did not focus on interbrain dynamics during social interactions. More recent studies have adopted innovative methodologies such as hyperscanning, portable electroencephalography (EEG), or wireless functional near-infrared spectroscopy (fNIRS), which can simultaneously record brain activity in all interacting individuals [95,96,97,98,99], making it possible to understand when and how neural processes become synchronised between two or more people [99,100]. These advanced approaches have revealed that brain activity in interacting individuals is synchronised during social engagement [96,101]. For example, an EEG study assessing neural coherence in a classroom of teenagers demonstrated that the degree of synchronisation of brain activity between students predicted both student engagement and social dynamics in the classroom [95]. An fNIRS study showed increased intercerebral activity synchronisation in the premotor cortex when two subjects engaged in a finger-tapping synchronisation task compared to a control task without interaction [97]. Another fNIRS study showed greater synchronisation of intercerebral activity in the right superior frontal cortices in participants during a cooperative computer game compared to a competitive game [102]. This improved synchronisation was associated with better player performance. Other studies have shown intercerebral synchronisation in the frontotemporal region and temporoparietal junction when individuals are engaged in social interactions [95,103,104,105,106]. Finally, neurophysiological studies show that interbrain synchronisation involves motor regions and areas associated with attention, suggesting that this synchronisation depends in part on the mutual recognition of a partner’s role and actions and is associated with shared attention [42,91,103,107,108].

In sum, the discovery of mirror neurons has provided neural evidence for motor resonance in human and non-human primates. Although mirror neurons have been the subject of numerous studies and debates regarding their social function (see [89,109] for reviews), it is widely acknowledged that mirror neurons have a role in imitative processes, including behavioural synchronisation. Mirror neuron activation promotes motor resonance and intercerebral synchronisation, leading to interindividual behavioural synchronisation if the context of social interaction is appropriate: when individuals are affiliated, mutually attentive or share a common goal, for instance.

### 3.2. Do Dogs Possess Mirror Neurons? Drawing Insight from Mirror Neurons in Other Social Species

Thus far, mirror neurons have not been explicitly identified in dogs, but the evolutionary continuity of mammalian brains implies that mirror neurons may be present in all mammalian brains [110]. Indeed, brain areas involved in imitation and prosocial systems have been conserved over evolutionary time for many mammalian species [111]. This neurological conservation suggests that this behavioural mechanism is built upon an evolutionary ancient mechanism of motor resonance. Motor resonance would therefore be based on automatic action perception coupling in sensorimotor brain areas [112]. To date, in primate species and rodents, neurons encoding observed actions [113], emotions [114], and direction of attention [90] have been found in brain areas dedicated to the processing of self-related information [89]. For example, a recent study demonstrated that distinct neural populations in the prefrontal cortex in mice drive interbrain neural synchronisation during dyadic social interactions [115]. This interbrain neural synchronisation plays a role in behavioural synchronisation and maintaining social interaction. Another study showed that mirror neurons can be activated at the interspecific level in primates, as macaques recognise when their goal-directed actions are imitated by a human experimenter [116].

Moreover, additional experimental evidence demonstrates that mirror neurons are not only present in mammalian species, as they have been found in songbirds as well [117,118]. Researchers have found audiovocal neurons in birds with properties similar to those of mirror neurons in the premotor cortex in primates. These neurons exhibit audiovocal mirroring properties during the perception of auditory information. This homology suggests a phylogenetic origin of neurons related to the self and the actions of others, a solution conserved over evolution for imitation and social learning in social species (see Figure 3) [89].

This set of evidence strongly suggests that dogs also possess mirror neurons. Given the neurological conservation of the mirror system and the cognitive abilities of dogs toward social cues and human actions, it is conceivable that due to their extensive history of living alongside humans, dogs have developed cerebral representations of actions akin to those of humans, resulting in a set of representations of actions common to both dogs and humans [119]. Consequently, it becomes entirely plausible that human mirror neurons, on the one hand, and canine mirror neurons, on the other, could be activated during dog–human interactions involving behavioural alignment. Interspecific behavioural synchronisation would then be based on interspecific motor resonance. Neurophysiological and behavioural studies are needed, however, to substantiate this hypothesis.

## 4. Evidence for Motor Resonance from Behavioural Studies

### 4.1. Motor Resonance and Motor Contagion

In humans, when the automatic response triggered by motor resonance is not inhibited, it results in the execution of the corresponding action by the observer (see Figure 1). There is thus a spread of behaviours from the observed person to the observer [120]. This is known as motor contagion: a behaviour is propagated from one individual to another through sensorimotor entrainment [121]. Motor contagion leads to behavioural synchronisation if the observed individuals continue to exhibit their behaviour, resulting in interacting individuals performing the same behaviour simultaneously. Alternatively, it leads to rapid mimicry if the observed individuals cease their behaviour and are quickly mimicked by the observers [109]. Motor contagion is therefore the behavioural result of motor resonance in species possessing a mirror neuron system [122]. Demonstrating motor contagion within the context of dog–human interactions would provide supporting evidence for the existence of interspecific motor resonance. In dogs, the spread of behaviours can be observed at the intraspecific level during play interactions, when dogs produce play solicitation behaviours, lowering the front of their body toward the ground and opening and relaxing their mouth; these behaviours are then quickly copied by their conspecifics [3]. At the interspecific level, numerous studies have examined the spread of behaviours in dogs during their interactions with humans. We will present two examples: contagious yawning, a case of motor contagion leading to rapid mimicry, and contagious locomotor behaviours, a case of motor contagion leading to behavioural synchronisation.

The contagion of yawning has been the subject of many studies (see [123] for a review) as it is a classic example of motor contagion. In human and non-human primates, the perception of yawning via visual and/or auditory cues triggers yawning in the observer [5,124,125,126,127] through the activation of the mirror neuron system [128]. Contagious yawning has also been studied at the interspecific level between chimpanzees and humans and between dogs and humans. The perception of human yawning has been shown to induce yawning in young chimpanzees but not in infant chimpanzees [129,130]. Similarly, contagious yawning has not been found in puppies when they perceive human yawning [131]. Some studies on adult dogs have reported that yawning is contagious when dogs visually and/or auditorily perceive familiar or unfamiliar human yawning [5,132,133,134]. However, other studies have found different results indicating the absence of contagious yawning when dogs perceive human yawning [135,136]. These discrepancies may stem from the fact that the function of yawning is not clearly established [137,138]. Yawning may have a different social function in humans and dogs, which could explain the variability in study results.

Other studies have highlighted locomotor contagion during dog–human walks. Dogs and humans have been walking together for millennia, and there is no ambiguity about the function of walking behaviour: to move from one location to another. Studies on dog locomotor synchronisation with humans have shown that locomotor entrainment occurs when dogs perceive humans walking [29,30,31,32]. The results of these studies indicate that when a dog’s owner starts moving, this induces the dog to move within a few seconds (see Figure 4). This locomotor contagion occurs whether the dog is in an indoor or outdoor space, known or unknown to the dog, and whether the owner is alone or in a group of people (see Figure 4). This locomotor contagion then enables dogs to synchronise their locomotion with their owners: when owners change their walking pace, dogs adapt their own pace within about 2 s. In humans, locomotor contagion relies on motor resonance and the activation of mirror neurons [46,139,140,141]. Individuals then synchronise their locomotion to reduce discrepancies between representations of the observed motor behaviour and their own behaviour [142]. Locomotor contagion in dog–human interactions could conceivably rely on a similar mechanism, namely interspecific motor resonance.

Our hypothesis proposes that the locomotor contagion from humans to dogs arises from an interspecific motor resonance mechanism. However, it is crucial to acknowledge the possibility that this phenomenon might also result from a process that does not necessarily involve the activation of canine mirror neurons during the perception of human actions. While it is highly likely that dogs possess mirror neurons, it remains possible that these neurons do not activate in interspecific interactions. The dogs’ perception of human action would then not trigger the canine motor areas associated with that action but rather other cerebral areas, such as cerebral reward circuits, prompting dogs to mimic humans without relying on sensorimotor entrainment. The contagion of locomotor behaviours would then stem from an associative learning process [143]. Indeed, dogs quickly form stimulus–reward associations [144], and human cues can be reinforcers for dogs [145], so dogs may walk with their owner only because it is rewarding for them. Neurophysiological studies are thus essential to validate or refute our hypothesis of interspecific motor resonance. Investigating the areas activated in dogs’ brains when perceiving human behaviours, such as walking, and comparing them to those activated during the dogs’ own walking could unveil whether the locomotor synchronisation between dogs and humans is indeed a result of interspecific motor resonance or if alternative mechanisms are at play.

### 4.2. Motor Interference and Motor Inhibition

The perception of an action triggers the activation of representations related to that action, leading to motor resonance. Motor resonance can be demonstrated by studying the effect of action perception on the execution of another action [146,147,148]. Studies in humans have shown that observing an action can interfere with the simultaneous execution of another action [41,149,150,151]. More specifically, when a subject is instructed to perform an action while observing another individual performing the same action, the execution of the action is facilitated by motor resonance. However, if the observed action is incongruent with the performed action, this disrupts the execution of the action; this is known as motor interference [121,151]. For example, if an individual performs a hand movement in a consistent direction, either up and down or from left to right, the trajectory of the individual’s hand movement is less rectilinear if the individual is observing someone else performing a gesture in the incongruent direction compared with a gesture in the same direction [151]. Similarly, when opening or closing a fist, the subject’s reaction time to perform the action is delayed if the subject observes an incongruent movement compared with a congruent one [152]. In fact, the perception of the incongruent action activates the representations associated with this action simultaneously with the representations associated with the action that the subject is performing. This results in a disruption in the execution of the action due to the automatic tendency to imitate caused by the activation of mirror neurons [121]. However, subjects are still able to perform their actions, despite being influenced by the perceived action, as the automatic response triggered by motor resonance can be inhibited (see Figure 1). In other words, when observers perceive an action, they first exhibit an automatic tendency to imitate it, and then their behavioural response is to copy the action, avoid copying it, or perform a complementary action, depending on the context of the social interaction.

Thus far, motor interference and inhibition of motor resonance have only been studied in humans, and to our knowledge, there are no publications on this topic involving dogs. Studies are therefore needed to determine whether such properties are present when dogs perceive the actions of their conspecifics or humans. These studies will need to focus on actions that are part of the canine behavioural repertoire. If the perceived action is not within the observer’s behavioural repertoire, sensorimotor representations associated with the action cannot be present, and consequently, motor resonance cannot occur [50].

### 4.3. Social Modulation of Motor Contagion

Motor contagion does not occur all the time or in all contexts. As a matter of fact, it is accentuated or inhibited by certain social modulators, such as the level of familiarity between interacting individuals, the number of social agents, attentional state, and motor experience (see [2] for a review). The social modulation of motor contagion has been the subject of numerous behavioural studies in humans. Studies have also been carried out on modulation at the interspecific level between dogs and humans. Firstly, familiarity with the model has been shown to have an effect on motor contagion and behavioural synchronisation. It is well established in humans that more affiliated individuals exhibit greater intercerebral and behavioural synchronisation compared to unfamiliar individuals [46]. The effect of familiarity is also present in dogs, whether with conspecifics or humans: motor contagion is more frequent between familiar dogs than between unfamiliar dogs [3,153]. Studies at the interspecific level have shown that pet dogs synchronise more with their owners than shelter dogs do with their caregivers [29,31], and the relationship between pet dogs and their owners is usually stronger than that between shelter dogs and their caregivers. Behavioural synchronisation thus acts as a social glue, strengthening bonds between familiar individuals [154]. Another study found that dogs exhibit a social preference for unknown humans who synchronise with them, which is further proof that behavioural synchronisation acts as a social glue at the intraspecific and interspecific levels [155]. Another social modulator of motor contagion is the number of individuals involved in the interaction. In humans, as the number of individuals increases, motor resonance becomes stronger, resulting in more pronounced behavioural synchronisation [47,156,157,158]. Observing multiple agents thus further activates motor representations associated with the perceived action [47]. In the context of dog–human interactions, a study investigated the effect of the number of humans on dogs’ locomotor synchronisation with humans [32]. The results indicated that the level of dogs’ synchronisation with their owner was equivalent, whether the owner was alone or with two other people. However, dogs’ visual attention towards humans increased when the humans were in a group compared to when the owner was alone. We note that the number of people in the group may not have been sufficient to observe an effect on the dogs’ locomotor synchronisation. Further studies are therefore needed to explore the effect of the number of humans more comprehensively.

Locomotor synchronisation in humans is modulated by the number of individuals but also by their locomotor profile [159]. Specifically, an individual’s movement speed profile modulates the degree of behavioural synchronisation [159]. Individuals with different speed profiles synchronise less than individuals with similar speed profiles, but they still adapt to each other to synchronise [160]. A study explored the impact of locomotor profiles on dogs’ locomotor synchronisation with humans by investigating dogs’ locomotor synchronisation with children [161]. Children have an immature locomotor pattern [162], yet the study showed that dogs were capable of synchronising with children’s locomotion. This research could be further developed by studying the effect of other human locomotor patterns on dogs’ locomotor synchronisation, such as the locomotion of elderly people. Locomotor patterns of elderly people are slowed down due to age-related decline in perceptual and locomotor abilities [162]. This research would provide a better understanding of how human locomotion influences dogs’ perception and motor response.

Another modulator of motor contagion in humans is motor experience [163]. As the individual’s level of expertise increases, motor resonance and consequently behavioural synchronisation become stronger [89]. This arises from the fact that visual and motor representations of action become linked through the observation and execution of the same action [164,165]. Studies have shown that brain activity during the perception of an action is higher in humans with experience in that action than in individuals with no expertise in the action [166,167]. Motor experience is acquired very early in humans, as babies just a few weeks old are already capable of behavioural synchronisation with adults [168]. Behavioural synchronisation also occurs very early in dogs, as a study demonstrated that one- and two-month-old puppies are able to synchronise with an adult person, whether familiar or not [169]. Further studies are needed to establish the effect of expertise on adult dogs’ behavioural synchronisation with humans.

Other social modulators of behavioural synchronisation, such as the spatial configuration of individuals and the level of attention, have not yet been studied in dogs, leaving room for future research to substantiate the social modulation of interspecific behavioural synchronisation.

## 5. Conclusions

Dogs are widely acknowledged to exhibit behavioural synchronisation with humans. Behavioural synchronisation in humans is based on motor resonance and the activation of mirror neurons. The literature reveals that, given dogs’ social cognitive abilities, they possess the prerequisites for motor resonance. Despite the absence of neurophysiological studies on the neural substrate of behavioural synchronisation in dogs, research on other species suggests that a mirror neuron system may exist in all social mammals. Neurological studies on dogs would confirm the existence of mirror neurons in dogs, providing direct evidence for the existence of interspecific motor resonance. Motor resonance leads to motor contagion, which can be modulated by various social factors. Behavioural studies indicate that motor contagion and its social modulation in humans are also present in dogs, thus reinforcing the hypothesis of interspecific motor resonance. However, in light of these behavioural studies, it could also be assumed that dogs’ behavioural synchronisation with humans is based on an associative learning mechanism. Additional studies are therefore necessary to determine whether the properties of motor resonance, notably motor interference, are present in dog–human interactions. This review thus paves the way for future behavioural and neurophysiological research to confirm whether interspecific motor resonance underlies dogs’ behavioural synchronisation with humans.

## Figures and Tables

**Figure 1 animals-14-00548-f001:**
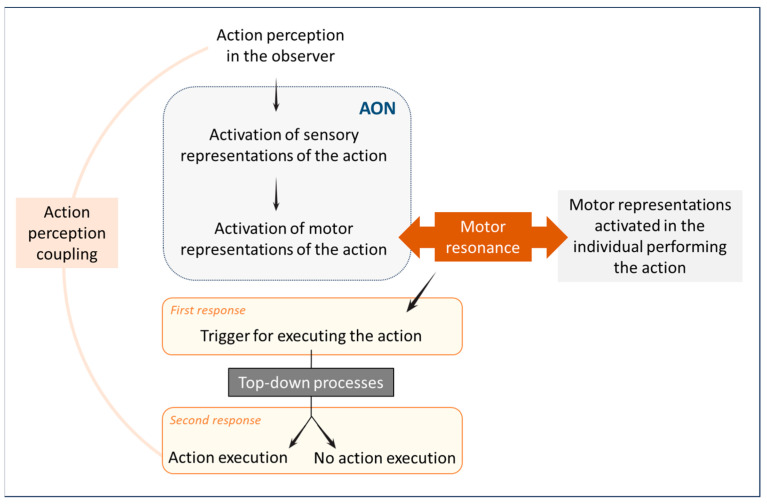
Sensorimotor processes associated with motor resonance in humans. The perception of an action activates the observer’s sensory and motor representations associated with that action. Motor resonance induces a first response, which is a trigger for executing the action. This initial response is modulated through top-down processes, either reinforcing or inhibiting it. When this initial response is uninhibited, it leads to the observer executing the perceived action.

**Figure 2 animals-14-00548-f002:**
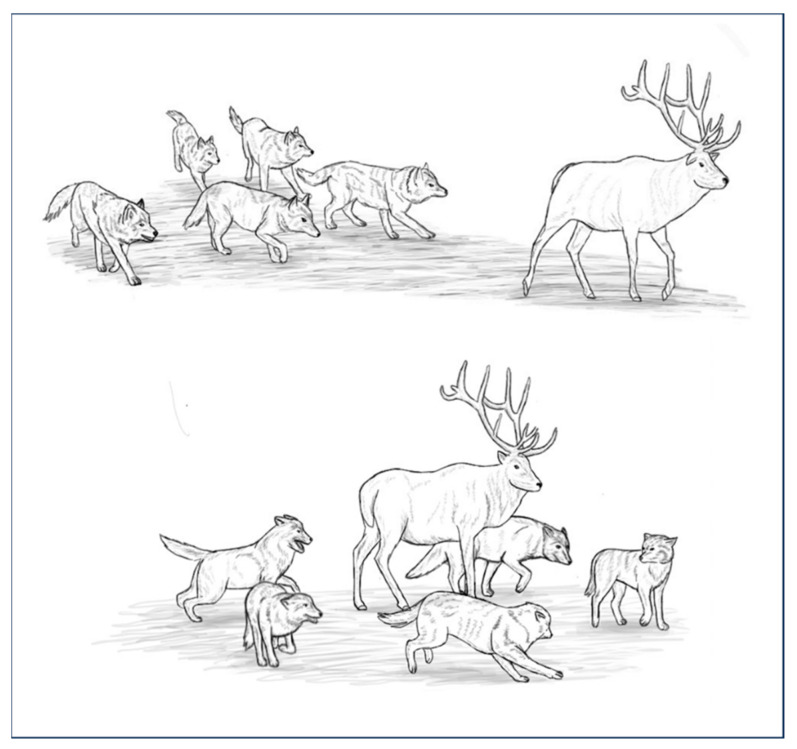
Canine behaviours while hunting.

**Figure 3 animals-14-00548-f003:**
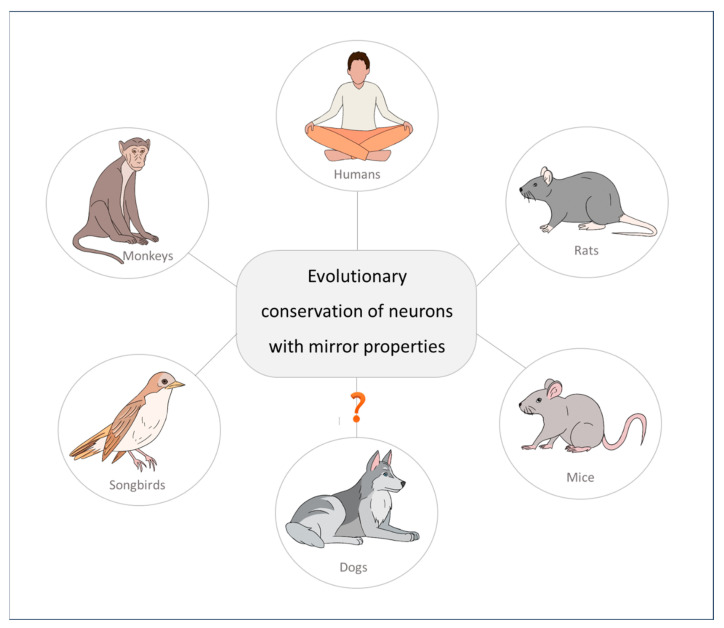
Mirror neurons, a neurological solution conserved over evolution for behavioural alignment in social species.

**Figure 4 animals-14-00548-f004:**
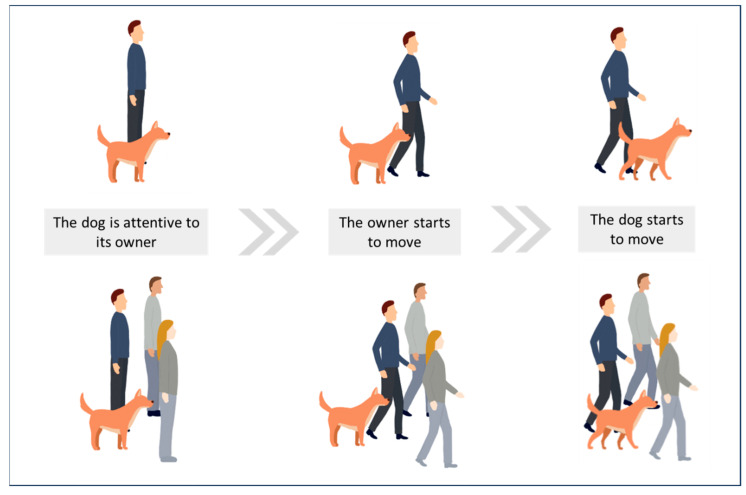
Interspecific locomotor synchronisation.

## Data Availability

No new data were created or analysed in this study. Data sharing is not applicable to this article.

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
