# Peer review of "Behavioural Synchronisation between Dogs and Humans: Unveiling Interspecific Motor Resonance?"

_animals, 2024, doi:10.3390/ani14040548_

Round 1

Reviewer 1 Report

Comments and Suggestions for Authors

This paper presents findings from studies on the mechanisms underlying behavioral synchronization in humans to examine whether similar mechanisms might underly interspecific behavioral synchronization between dogs and humans. The review is comprehensive, extremely well-written and organized. My background is in dog behavior-not cognition or neurobiology-yet the paper is written in such a manner that it is understandable by readers in other fields. Additional strengths include the identification of gaps in existing research, suggestions for future research, and nicely done figures. I have only a few suggestions that might improve the paper.

Simply Summary: This summary is for the public, so I suggest including brief definitions for motor resonance and mirror neurons in line 12.

Lines 153-162: In this section where you detail canid hunting tactics, it would be great to include a citation or two on cooperative hunting in African Wild Dogs (Lycaon pictus), an extremely social species. Possible recent papers include:

Hubel et al. 2016. Additive opportunistic capture explains group hunting benefits in African wild dogs. Nature Communications. doi:10.1038/ncomms11033. (note this paper also reviews hunting tactics in wide open areas)

Walker et a. 2017. Sneeze to leave: African wild dogs (Lycaon pictus) use variable quorum

thresholds facilitated by sneezes in collective decisions. Proc. R. Soc. B 284: 20170347.

doi:10.1098/rspb.2017.0347

Line 159: I suggest changing “canine”, which refers only to domestic dogs to “canid”, which refers to members of Family Canidae, given that reference 44 concerns and Figure 2 shows wolves.

Conclusion and future directions: I was a bit surprised to see the suggestion that dogs’ behavioral synchronization with humans might be based on associative learning mechanisms (rather than interspecific motor resonance) in this last section of the paper because associative learning mechanisms are not presented as an alternative explanation earlier in the paper. Consider adding a brief paragraph on these alternative mechanisms earlier in the manuscript.

Minor point: There are some places in the pdf where the font size is much larger (lines 45-49; lines 149-150; lines 268-271).

Reviewer 2 Report

Comments and Suggestions for Authors

What is not mentioned at all in the manuscript is that not all the behaviours are mimicked. The motor synchronization does not happen always and for all the actions. For example, to demonstrate the existence of unconscious and/or rapid mimicry you have to compare the stimulus under study with a control stimulus. Mimicry and synchronization have an adaptive value that authors did not mention in the introduction, and I think is the most important section that have to be mentioned. Why has evolution favored these phenomena?

Line 49: I don’t understand the need for specifying that dogs descend from predators, it is misleading and out of context.

Line 50-56: there are several recent studies about domestication that authors did not mention. Maybe it could be useful for authors to read also other more recent paper (e.g. Perri et al. 2021, Bergstrom et al., 2020, Greig et al. 2015)

Line 93-144:  the title is “the case of humans” but, except for the last sentences, I think it is a general explanation of the phenomenon and not something really related to humans. Humans spontaneously mimic others’ action in lots of different ways that are not mentioned: for example, in Rapid Facial Mimicry (already mentioned for dogs in the introduction), or leg shaking, spontaneously imitating vocal accent and, even more complex, mimicking the use of objects such as cigarettes or smartphones.

I understand that the literature about humans is very huge and maybe it is not so necessarily, but if authors want to give an overview about humans maybe they should give some more examples and not only the general definition of the phenomenon.

Line 123- 124: What does “If the most appropriate behavioural response in the context of the interaction is an imitative one, then motor resonance is not inhibited, resulting in the observer performing the behaviour”? All the behaviours can potentially evoked a replication but some imitation are inhibited?? How/who chose the “most appropriate behavioural response” in unconscious imitation? Moreover, authors are using here, for the first time, the term “imitation”, that have different meaning (the same of copying and not the same of mimicry). Same concern in the line 304.

Line 148-150: Like all social species, dogs have the ability to discriminate the behaviours of their conspecifics and take this information into account to adjust their own behaviours.

The cited literature (41, 42) is about conscious imitation. (Same concerns already stated)

Line 150-162: This section is completely misleading, inaccurate and does not match with the purpose of the manuscript.

Line 165-166: This is not true. There is a plethora of papers (both domestic and wild species) demonstrating that other species can discriminate humans’ expressions of emotions and understand some human behaviours. From non-human primates to horses and others.

Line 169: dogs adopt humans’ behaviours?

Line 220: LIP is the abbreviation of?

Line 270-271: This sentence is completely out of context. Moreover, why authors always talk bout “ancestors of dogs”…. The ancestor of dogs is the wolves (in fact, authors are citing wolves’ literature)

Line 271-287: Authors are talking about mammals in general, and not dogs… Maybe the title of the paragraph should be changed.

Line 289-291: I do not understand. Authors talked about primates, then dogs, then mammals (and birds), and then primates again?

Line 290-293: This could be true for dogs, but not for humans. Dogs derive from a domestication process and a process of artificial selection that have been lasting for thousands of years, driving through the separation from dogs and wolves and then to all the different dog breeds we know today. As authors said in their introduction, dogs were selected for their capability in following/understanding human actions. This is something different from saying that DOGS AND HUMANS HAVE DEVELOPED A SHARED RAPRESENTATION OF ACTIONS OVER THE COURSE OF THEIR COHABITATION. All dogs descend from the first domesticated wolf, all dogs are the product of the artificial selection operated by humans. Not the contrary. Not all humans lived with /live with /know/understand dogs.

Line 294-296: As it is written, this sentence implies that all dog-human interaction activates mirror neurons.

Line 313: The behaviours authors are talking about are not copied, but rapidly mimicked (Rapid mimicry and Rapid Facial Mimicry).

Line 316: I do not understand. Authors mention yawning. Contagious yawning is a motor resonance phenomenon, but what does it have to do with motor synchronizations defined as authors wrote in the introduction? In contagious yawning, the latency can also be 1 or 2 minutes. So, not the “same behaviour” at the “same time”. I am confused and still do not understand the focus of the manuscript.

Line 317-326: If a dog is more attentive and tense after seeing after recognizing a negative emotion, this is not linked with behavioural synchronization. I am not able to catch the point of the entire paragraph.

Yawning is linked to behavioural synchronization in lions, if authors want to give an overview on other social carnivores. But maybe they could start from yawning in wolves.

Line 342: I am not sure about why authors talk about walking together. Do they think this activity is linked with motor resonance? Really? Do they think it is something unconscious?

Round 2

Reviewer 1 Report

Comments and Suggestions for Authors

Thank you for addressing my comments and those of Reviewer 2. The manuscript is substantially improved and I have no additional suggestions.

Author Response

Thank you sincerely for taking the time to review our manuscript and for your invaluable feedback. We appreciate your thorough evaluation and positive assessment of our article.

Reviewer 2 Report

Comments and Suggestions for Authors

I would like to thank the authors for the responses they gave to my concerns.

I think that now the manuscript has improved. 

I still have some concern about line 197: "dogs perceive and copy some human behaviours". Which behaviours? As it is written, this sentence is not very clear, please give some examples.

Author Response

Thank you very much for your time and for your constructive comments, which greatly contributed to enhance the overall quality of our manuscript.

Regarding your concern about line 197, we understand the need for clarification. To address this, we have detailed some examples, line 198-207: “For instance, studies using the "Do as I do" procedure showed that dogs learned to copy actions demonstrated by humans such as turning around or jumping over an obstacle, and then could generalize it to other human behaviours they hadn't specifically been taught, like opening a door or moving an object [64,65]. Dogs were also found to copy human actions even when they were not causally relevant: dogs copied their owner pushing a door leading to a food reward, but they also copied their owner touching colored dots next to the door, even though it did not give them access to the food [20]. Lastly, another study showed that when faced with a stranger, dogs copied their owner's behaviour, by approaching the person or staying back based on their owner’s behaviour [66].”

We hope this is clearer now, and we remain available for any further suggestions or clarifications you may require.